# Schistosome infection among pregnant women in the rural highlands of Madagascar: A cross-sectional study calling for public health interventions in vulnerable populations

Raphäel Rakotozandrindrainy[1], Rivo Andry Rakotoarivelo[2], Irina Kislaya[3,4]*, Valentina Marchese[3,4]*, Tahimandranto Rasamoelina[5], Jeannine Solonirina[1], Elveric Fesia Ratiaharison[2], Ravo Razafindrakoto[5], Nantenaina Matthieu Razafindralava[5], Njary Rakotozandrindrainy[1], Mickael Radomanana[2], Mala Rakoto Andrianarivelo[5], Philipp Klein[3,4], Eva Lorenz[3,4], Anna Jaeger[3,4], Pytsje T. Hoekstra[6], Paul L. A. M. Corstjens[6], Norbert Georg Schwarz[3], Govert J. van Dam[6], Jürgen May[3,4], Daniela Fusco[3,4]*, on behalf of the freeBILy consortium[¶]

1 University Antananarivo, Antananarivo, Madagascar, 2 Infectious diseases service, University Hospital Tambohobe, Fianarantsoa, Madagascar, 3 Department of Infectious Diseases Epidemiology, Bernhard Nocht Institute for Tropical Medicine (BNITM), Hamburg, Germany, 4 German Center for Infection Research (DZIF), Hamburg-Borstel-Lübeck-Riems, Germany, 5 Centre d'Infectiologie Charles Mérieux (CICM), Antananarivo, Madagascar, 6 Leiden University Medical Center, Leiden, The Netherlands

☯ These authors contributed equally to this work.
¶ Membership of the freeBILy consortium is listed in the Acknowledgments.
* irina.kislaya@bnitm.de (IK); valentina.marchese@bnitm.de (VM); fusco@bnitm.de (DF)

## Abstract

### Introduction

Schistosomiasis is a parasitic infection highly prevalent in sub-Saharan Africa (SSA) with Madagascar being among the countries with highest burden of the disease worldwide. Despite WHO recommendations, suggesting treatment of pregnant women after the first trimester, this group is still excluded from Mass Drug Administration programs. Our study, had the objective to measure the prevalence of schistosome infection among pregnant women in Madagascar in order to inform public health policies for treatment in this vulnerable population.

### Methods

Women were recruited for this cross-sectional study between April 2019 and February 2020 when attending Antenatal Care Services (ANCs) at one of 42 included Primary Health Care Centers. The urine-based upconverting reporter particle, lateral flow (UCP-LF) test detecting circulating anodic antigen was used for the detection of schistosome infections. To identify factors associated with the prevalence of schistosome infection crude and adjusted prevalence ratios and 95% CIs were estimated using mixed-effect Poisson regression.

### Results

Among 4,448 participating women aged between 16 and 47 years, the majority (70.4%, 38 n = 3,133) resided in rural settings. Overall, the prevalence of schistosome infection was

**Data Availability Statement:** All relevant data are within the paper and its Supporting Information files.

**Funding:** This publication was produced by freeBILy which is part of the EDCTP2 programme supported by the European Union (grant number RIA2016MC-1626-FREEBILY to NGS, RR, RAR, TR, DF). The views and opinions of authors expressed herein do not necessarily state or reflect those of EDCTP. Additional funding support came from the German Centre for Infection research (DZIF) through the projects SCHISDIMA (project number: TI 03.907 to NGS, JM) and NAMASTE (project number: 8008803819 to DF- salaried on the project, JM). The funders had no role in study design, data collection and analysis, decision to publish, or preparation of the manuscript.

**Competing interests:** The authors have declared that no competing interests exist.

55.9% (n = 2486, CI 95%: 53.3–58.5). A statistically significant association was found with age group (increased prevalence in 31–47 years old, compared to 16–20 years old (aPR = 1.15, CI 95%: 1.02–1.29) and with uptake of antimalaria preventive treatment (decreased prevalence, aPR = 0.85, CI 95%: 0.77–0.95). No other associations of any personal characteristics or contextual factors with schistosome infection were found in our multivariate regression analysis.

## Discussion and conclusion

The high prevalence of schistosome infection in pregnant women supports the consideration of preventive schistosomiasis treatment in ANCs of the Malagasy highlands. We strongly advocate for adapting schistosomiasis programs in highly endemic contexts. This, would contribute to both the WHO and SDGs agendas overall to improving the well-being of women and consequently breaking the vicious cycle of poverty perpetuated by schistosomiasis.

### Author summary

Schistosomiasis is a parasitic infection highly prevalent in sub-Saharan Africa and in Madagascar, where pregnant women are systematically excluded from prevention and control strategies. Our study shows the urgency for adapting public health strategies. We report a high prevalence of schistosome infection among pregnant women in Madagascar. The lack of systematic treatment for this vulnerable group can have a direct impact on the well-being of women and their offspring contributing to the vicious cycle of poverty perpetuated by schistosomiasis.

## Introduction

Women's health is among the priorities of the Sustainable Development Goals (SDGs) since it has been shown that the well-being of women has a direct impact on other people, societies and the countries where they live [1]. Pregnancy, is considered a critical healthcare need of women, and the World Health Organization (WHO) states that "Women who remain healthy during pregnancy and after birth are more likely to stay healthy later in life and have better birth outcomes, influencing infancy, childhood and adulthood" [1].

Globally, there are over 213 million pregnancies every year, of which an estimated 190 million (89%) occur in low-resource settings where the risk of poor birth outcomes is highest due to multifactorial risk factors including parasitic infections during pregnancy [2,3].

Schistosomiasis is a parasitic infection highly prevalent in SSA where more than 90% of those requiring treatment live [4]. It is caused by different species of the trematode schistosome, of which *S. mansoni* and *S. haematobium* are the most frequent worldwide. Since the 2001 World Health Resolution calling for widespread treatment for schistosomiasis [5], the disease is receiving unprecedent attention from the global health community which targets the elimination of schistosomiasis as a public health problem by 2030 through the recently released WHO 2021–2030 road map for Neglected Tropical Diseases (NTDs) [6]. To boost this goal, *ad-hoc* guidelines for schistosomiasis were launched in 2022 by the WHO where, on the basis

of six specific recommendations, new strategies for the management of the disease are suggested [7].

Historically, schistosomiasis control strategies in highly endemic areas heavily relied on preventive chemotherapy through mass drug administration (MDA) with 40 mg/kg praziquantel (PZQ) for school-aged children [8]. The recent WHO guidelines suggest treatment for all individuals above 2 years of age in highly endemic areas, including pregnant women after the first trimester, through the specific recommendation number one [7]. Despite this recommendation, and the growing body of evidence on poor pregnancy outcomes and lifelong effects on child's health associated with schistosome infection [9,10], pregnant women are still excluded from MDA programs [11]. However, the scarcity of data on treatment toxicity during pregnancy [12] still poses a barrier for MDA program expansion.

Among the affected countries, Madagascar has the fifth largest burden of schistosomiasis worldwide, with 106 out of 113 districts considered endemic for the disease. In the country, the two most frequent species of *Schistosoma* co-exist with a distinct geographical segregation locating *S. haematobium* in the western and northern regions of the island, while *S. mansoni* in the eastern and southern parts so as in the central highlands of the country [13]. MDA campaigns are regularly offered in the country but pregnant women are not yet targeted for treatment. Even if national guidelines allow schistosomiasis treatment for pregnant women laboratory diagnosed for the disease [13], in absence of integrated programs, i.e. through antenatal care (ANC) services, the treatment is based on an out-of-pocket payment leading *de facto* to no treatment for this vulnerable group. Successful examples of ANC-integrated chemoprophylaxis for parasitic diseases come from the Intermittent Preventive Treatment of Malaria for Pregnant Women (IPTp) that is being adopted for more than 30 years in endemic countries with solid data showing the impact on both women's and children's health [14,15].

The aim of the study was to provide further evidence to support the adoption of public health strategies targeted at pregnant women for the management and control of schistosomiasis in endemic areas. The objectives of the study were to estimate the prevalence of schistosome infection and identify factors associated with the infection among pregnant women in the rural highlands of Madagascar.

## Methods

### Ethics statement

The freeBILy trial was implemented according to ICH-GCP guidelines in compliance with the Helsinki Declaration and other legal and ethical requirements. The study protocol received approval from the National Ethics Committee of Madagascar (ref. no 022-SANP/CERBM of 05/03/2018) and the Ethics Committee of the Hamburg State Medical Chamber in Germany (ref. no PV5966 of 18/03/2019). Written informed consent was obtained from all participants or their legal guardians (under 18 years old). Study participation was voluntary, participants had a right to withdraw from the study at any time. The reimbursement to travel from the residence to the PHCCs was the only financial incentive offered for participation.

### Study design and target population

This cross-sectional study was conducted using baseline data on pregnant women enrolled in the cluster randomized controlled trial freeBILy (fast and reliable easy-to-use-diagnostics for eliminating bilharzia in young children and mothers) [16] (Pan-African Clinical Trial Register PACTR201905784271304), in which women, in the intervention arm, were treated with PZQ (40mg/kg) on site for schistosome infection following a Point-of-care Circulating Cathodic Antigen (POC-CCA) screening. The POC-CCA results were not used for the analysis of this

study since for the purpose of the present study a more accurate and precise test was selected to describe prevalence [17].

Women aged 16 years old or more, with single or twin pregnancy, were recruited at routine ANC services during the second or third pregnancy trimester between April 2019 and February 2020 at one of 42 included Primary Health Care Centers (PHCCs) located in the Itasy, Bongolava, and Amoron'I Mania regions of Madagascar, both known to be endemic for *S. mansoni*. Excluded from the study were women having a history of congenital anaemia or blood transfusion (in order to exclude confounding factors for the primary outcome of the clinical trial freeBILy), epileptic or convulsive episodes (to prevent side effects after treatment with Praziquantel), fever at the day of recruitment (in order to limit the burden of possible overtreatment during pregnancy), as well as those who withdrew informed consent or did not have the urine sample for laboratory analysis.

### Data and sample collection

Sociodemographic information, data on health status, previous clinical history and current medication intake were collected by trained health professionals at PHCCs using paper-based case report forms. Data was digitalized in REDCap [18] on the basis of a double data entry procedure and checked for completeness and consistency by a data quality manager.

A urine sample was collected from each participant and stored on-site at room temperature for a maximum of seven days until transportation to the central laboratory, in order to decrease the deterioration of the samples. At reception at the central laboratory (*Centre d'infectiologie Charles Merieux*–CICM, Antananarivo), samples were aliquoted for long-term storage at -80˚C [19].

### Laboratory testing

The upconverting reporter particle, lateral flow test detecting circulating anodic antigen (UCP-LF CAA) was used for the detection of the *Schistosoma* genus specific CAA antigen as marker of active infections, as described previously [19]. All urine samples were tested using the UCAA*hT*417 format [20]. Samples with known CAA-levels were included in the analysis to validate the test as well to quantify the CAA concentration in each sample.

Samples were considered positive if the CAA concentration exceeded the cut-off of 2 pg/ml [20]. Post-testing quality control was performed by Leiden University Medical Center (LUMC), developer and producers of the test. Any invalid result was excluded from the final analysis.

### Statistical analysis

Participants' characteristics were summarized using frequencies and percentages for categorical variables and central tendency and dispersion statistics for numerical variables.

Prevalence of schistosome infection with respective 95% confidence intervals (CIs) was based on UCP-LF CAA test results and determined for the overall sample and stratified by participants' age, region of residence, urbanization, education level, working as a farmer, being the main contributor to the family income, ever being treated with PZQ, uptake of antimalaria preventive treatment and taking supplements during pregnancy.

To identify factors associated with the prevalence of schistosome infection crude and adjusted prevalence ratios (aPR) and 95% CIs were estimated using mixed-effect Poisson regression to account for the hierarchical structure of the data and variation across clusters (i.e. PHCCs) as a random effect.

Statistical analyses were performed using the software Stata for Windows, version 16 (Stata Corp, College Station, TX, USA). The significance level was set at 5%.

### Sensitivity analysis

We compared the sociodemographic characteristics of women with and without UCP-LF CAA test results using chi-square test. Following this comparison, we developed a sensitivity analysis, based on standard methods [21], to estimate the possible prevalence range and change in aPR estimates under two assumptions: 1) all participants without UCP-LF CAA test results considered negative for schistosome infection and 2) all participants without UCP-LF CAA test results considered positive. A difference of > 10% between the original estimate and the one obtained in sensitivity analysis was considered as an indication of bias [21].

## Results

### Participants' characteristics

From a total of 5,114 women recruited, 4,448 were included in the main analysis. The detailed participants' inclusion flowchart is shown in Fig 1.

Women included in the study were aged between 16 and 47 years, the mean gestational age was 24.7 weeks (SD = 1.9). The majority of women (70.4%, n = 3,133) resided in rural settings, 47.0% (n = 2,092) lived in the region of Amoron'iMania, 39.5% (n = 1,758) of Itasy, and the remaining 13.4% (n = 598) of Bongolava.

Of all women, 3.7% (n = 166) reported having no formal education, while 46.4% (n = 2,064) had secondary or higher education. Most of the participants were farmers (69.1%, n = 3,073), and 2.3% (n = 104) were the main contributors to the family income.

Regarding the clinical history, 11.4% (n = 498) of pregnant women reported being previously treated with PZQ while 35.3% (n = 1,565) reported having used other anti-helminth medication (albendazole), 44.1% (n = 1,962) mentioned uptake of iron or folate supplements during pregnancy, and 30.7% (n = 1,364) the antimalaria preventive treatment. A minority (1%, n = 45) reported having health insurance (Table 1).

### Prevalence of schistosome infection

Overall, the prevalence of schistosome infection among pregnant women of our sample was 55.9% (CI 95%: 53.3–58.5) (Table 2). Of those tested positive on site, 99.9% (n = 1801/1803) accepted treatment with PZQ.

The prevalence of schistosome infection increased with increasing age, ranging from 53.5% (CI 95%: 49.3–57.5) among women aged 16–20 years old to 61.8% (CI 95%: 58.1–65.3) among 30–47 years old. The proportion of positive for schistosome infection was 50.3% (CI 95%: 46.4–54.3) in Bongolava region, 56.1% (CI 95%: 52.1–60.0) in Amoron'i Mania and 57.6% (CI 95%: 53.7–61.3) in Itasy. Moreover, we report a 54.4% (CI 95%: 52.1–56.6) schistosome positivity among women living in rural settings and 59.5% (CI 95%: 53.2–65.6) in peri-urban settings.

Considering socioeconomic status, we estimated a higher prevalence of schistosome infection at 62.7% (CI 95%: 53.8–70.7) among those with no formal education compared to those with primary school 55.5% (CI 95%: 52.5–58.4) or secondary/higher education 55.8% (CI 95%: 52.5–59.0). Prevalence estimates were 54.4% (CI 95%: 52.0–56.9) among women who reported farming as their main occupation and 50.0% (CI 95%: 42.6–57.4) in the group that contributed mostly to the family income.

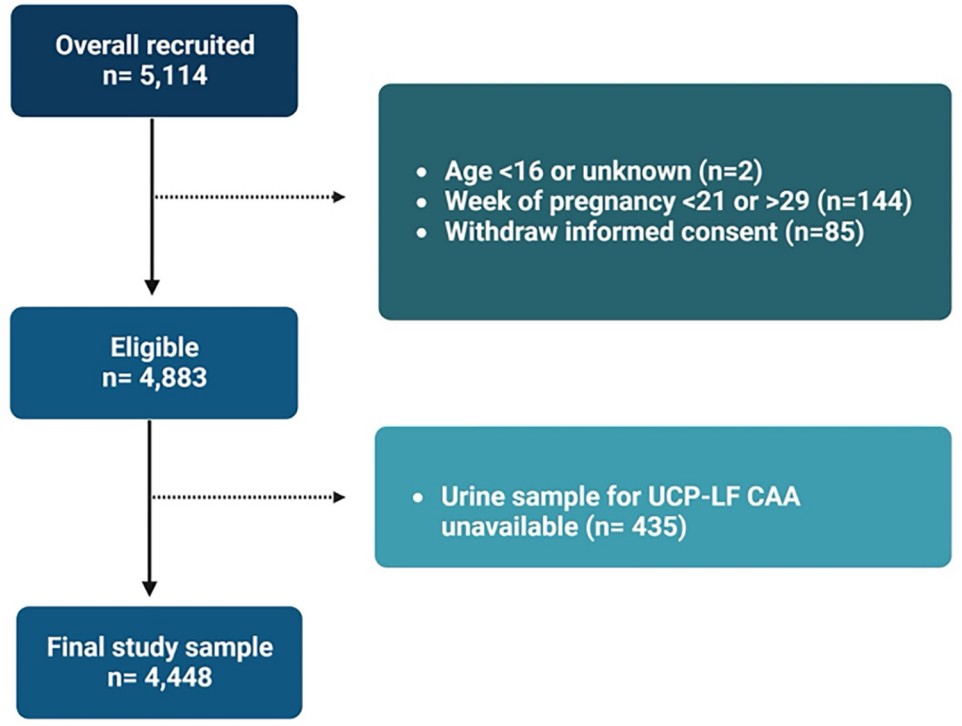

**Fig 1. Flowchart of participants' recruitment and urine sampling.**

Similar infection rates were estimated for women previously treated with PZQ and those without treatment (55.4% CI 95%: 49.4–61.3 vs. 56.1% CI 95%: 53.4–58.8) likewise infection rates stratified by another anti-helminth treatment with albendazole (56.0% CI 95%; 52.8–59.1 vs. 55.9% CI 95%; 52.5–59.2). A lower prevalence of schistosome infection was observed among those who reported uptake of antimalaria preventive treatment (51.3%, CI 95% 54.3–61.4 vs. 57.9%, CI 95% 47.1–55.6) (Table 2).

## Factors associated with schistosome infection

Associations between women's characteristics and schistosome infection status from the mixed-effect Poisson regression model are shown in Fig 2. After accounting for the effect of covariates, a statistically significant association was found with age group, indicating an increased prevalence in 31–47 years old, compared to 16–20 years old (aPR = 1.15, CI 95%: 1.02–1.29). Uptake of antimalaria preventive treatment (aPR = 0.85, CI 95%: 0.77–0.95), was associated with a decreased prevalence of schistosome infection. We observed no association between schistosome infection and other factors (region, urbanization, education, contribution to the family income, previous treatment with PZQ, uptake of iron/folate supplement and anti-helminth medication).

## Sensitivity analysis

We observed no differences in sociodemographic characteristics of women with and without UCP-LF CAA test results except in the distribution by the region of residence (S1 Table).

To assess potential bias in the reported estimates of the schistosome infection prevalence and the aPR due to the exclusion of 435 participants with missing UCP-LF CAA test results we

**Table 1. Sociodemographic characteristics and clinical history of participants.**

| Characteristics | n | % |
|---|---|---|
| **Age Group (n = 4,448)** | | |
| 16–20 | 1,364 | 30.7 |
| 21–25 | 1,344 | 30.2 |
| 26–30 | 937 | 21.1 |
| 31–47 | 803 | 18.1 |
| **Recruitment region (n = 4,448)** | | |
| Amoron'iMania | 2,092 | 47.0 |
| Bongolava | 598 | 13.4 |
| Itasy | 1,758 | 39.5 |
| **Urbanization (n = 4,448)** | | |
| Rural | 3,133 | 70.4 |
| Peri-urban | 1,315 | 29.6 |
| **Education (n = 4,448)** | | |
| No formal school | 166 | 3.7 |
| Primary | 2,218 | 49.9 |
| Secondary or higher | 2,064 | 46.4 |
| **Main contributor to family income (n = 4,448)** | | |
| Participant | 104 | 2.3 |
| Other | 4,344 | 97.7 |
| **Occupation (n = 4,446)** | | |
| Non-Farmer | 1,373 | 30.9 |
| Farmer | 3,073 | 69.1 |
| **Insurance (n = 4,448)** | | |
| No | 4,403 | 99.0 |
| Yes | 45 | 1.0 |
| **Ever treated with PZQ (n = 4,385)** | | |
| No | 3,887 | 88.6 |
| Yes | 498 | 11.4 |
| **Received anti-helminth medication (albendazole) (n = 4,436)** | | |
| No | 2,871 | 64.7 |
| Yes | 1,565 | 35.3 |
| **Received iron/folate supplement (n = 4,448)** | | |
| No | 2,486 | 55.9 |
| Yes | 1,962 | 44.1 |
| **Received antimalaria preventive treatment (n = 4,447)** | | |
| No | 3,083 | 69.3 |
| Yes | 1,364 | 30.7 |

performed a sensitivity analysis, as recommended in the literature [21]. Considering 435 participants with missing UCP-LF CAA test results as negative, we estimated the prevalence of schistosome infection of 50.9% (CI 95%: 48.5–53.4). Under the assumption of participants with missing UCP-LF CAA test results being all positive, the prevalence estimate was 59.8% (CI 95%: 57.3–62.3).

The aPR estimates were similar to ones from the main analysis under both assumptions, with the differences from the original estimates ranging from 0% to 6.3% (less than 10% for all variables).

**Table 2. Estimates of the prevalence of schistosome infection with respective 95% confidence intervals.**

| Characteristic | Prevalence | 95% CI |
|---|---|---|
| Overall | 55.9 | (53.3–58.5) |
| **Age Group** | | |
| 16–20 | 53.5 | (49.3–57.5) |
| 21–25 | 53.9 | (50.0–57.7) |
| 26–30 | 57.3 | (53.8–60.7) |
| 31–47 | 61.8 | (58.1–65.3) |
| **Recruitment region** | | |
| Amoron'iMania | 56.1 | (52.1–60.0) |
| Bongolava | 50.3 | (46.4–54.3) |
| Itasy | 57.6 | (53.7–61.3) |
| **Urbanization** | | |
| Rural | 54.4 | (52.1–56.6) |
| Peri-urban | 59.5 | (53.2–65.6) |
| **Education** | | |
| No formal school | 62.7 | (53.8–70.7) |
| Primary | 55.5 | (52.5–58.4) |
| Secondary or higher | 55.8 | (52.5–59.0) |
| **Main contributor to family income** | | |
| Participant | 50.0 | (42.6–57.4) |
| Other | 56.0 | (53.4–58.7) |
| **Occupation** | | |
| Non-Farmer | 58.1 | (55.6–62.8) |
| Farmer | 54.4 | (52.0–56.9) |
| **Ever treated with PZQ** | | |
| No | 56.1 | (53.4–58.8) |
| Yes | 55.4 | (49.4–61.3) |
| **Received anti-helminth medication (albendazole)** | | |
| No | 55.9 | (52.5–59.2) |
| Yes | 56.0 | (52.8–59.1) |
| **Received iron/folate supplement** | | |
| No | 56.2 | (52.4–60.0) |
| Yes | 55.5 | (52.6–58.3) |
| **Received antimalaria preventive treatment** | | |
| No | 57.9 | (54.3–61.4) |
| Yes | 51.3 | (47.1–55.6) |

## Discussion

This cross-sectional study reports a high prevalence (55.9%) of schistosome infections in pregnant women in the highlands of Madagascar. These findings are in line with a recent study conducted in Madagascar [22] in the adult population and reporting prevalence above 50% in different regions of the country. Our data show an alarming gap in the accomplishment of two main SDGs and WHO targets—the improvement of maternal health and the elimination of schistosomiasis as a public health problem—as pregnant women are still systematically excluded from schistosomiasis treatment in Madagascar as in many other endemic countries, despite WHO recommendations [7,11]. A recent meta-analysis reports an overall pooled prevalence of schistosomiasis in pregnant women of 13.2% in SSA, but also raised awareness about

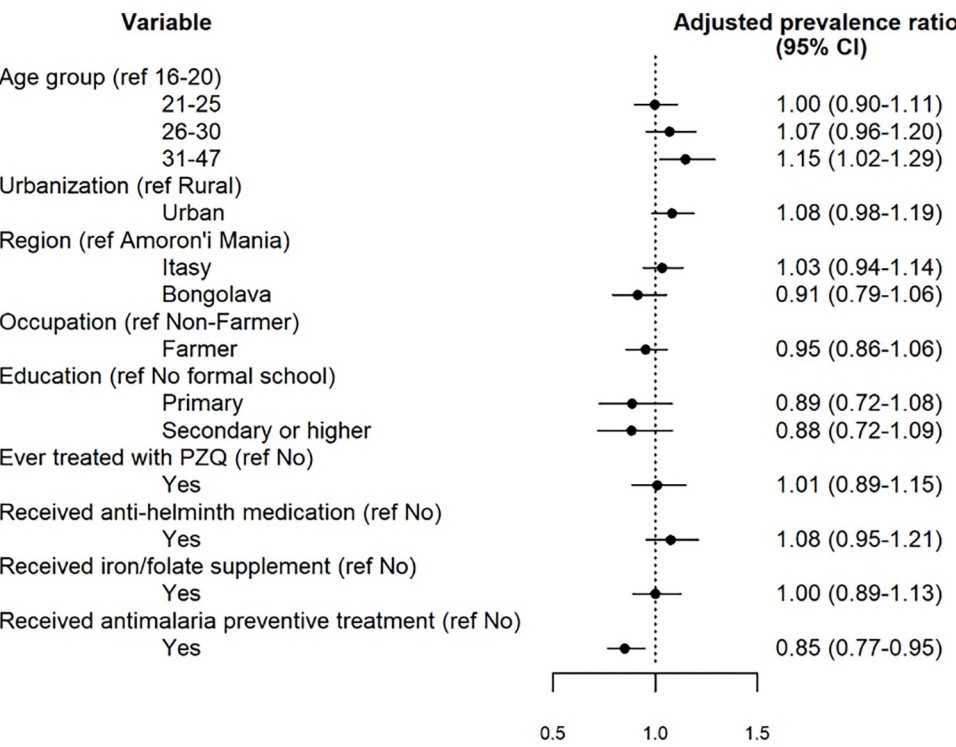

**Fig 2. Factors associated with schistosome infection.**

a lack of data on the determinants and outcomes of schistosomiasis during pregnancy [23]. Our study reports a high prevalence of schistosome infection among pregnant women (55.9%) in the highlands of Madagascar. To the best of our knowledge, this is the first study performed in the country on a wide territory spread over 42 different mainly rural communities in this specific population and the second [24] since the beginning of MDA in the country [25]. The previous study performed in pregnant women from samples collected in 2010, showed an overall prevalence of schistosome infection of 40.4% [24], however this was determined by a PCR-based detection method, and therefore a direct comparison with our CAA-based prevalence measurements is not appropriate. Additionally, our data show no relevant associations, apart from age and antimalaria preventive treatment, of any personal characteristics or contextual factors.

Considering that MDAs to control schistosomiasis started in Madagascar in 1999 [25], leastwise 82% of our study sample, who are younger than 30 years old, could have had access to PZQ treatment at least once in their lifetime. This aligns with the finding that women in the age group 31–47 (Fig 2) have a positive association with the infection. Similar findings have been described in the adult population of Madagascar [22], so as in other endemic countries [26–29], indicating that exposure to the parasite in adult life can have a critical role in the acquisition of the infection. Additionally, only the 11.4% reported to have received PZQ treatment in the past. Due to the self-reported nature of our question, we cannot exclude a reporting bias, nevertheless this data suggests that the coverage of MDA can be quite low even in those that attended school, which were almost all participants in our study. Additionally, the results of multivariable regression analysis showed that women previously treated with PZQ did not have a different chance to be actively infected with schistosomes. In this view, an active, structured and free of charge treatment in adult vulnerable populations should be considered

in highly endemic areas, since adults remain clearly at high risk of infection without a concrete possibility of accessing schistosomiasis treatment. In 2018 the WHO, in the absence of solid data on safety and tolerability of praziquantel during pregnancy, suggested to treat those pregnant women with a confirmed diagnosis of schistosomiasis [30].

At the current state of the art, test and treat strategies for schistosomiasis are still hard to implement in highly endemic countries due to the limited diagnostic options. The gold standard diagnostic test for schistosomiasis according to the WHO is microscopy that proved to be cost-effective and sufficiently performant in endemic areas [17,31]. Unfortunately, in order to have good sensitivity, microscopy must be performed on samples collected over three consecutive days [32]. Moreover, even though the technique is applicable to rural contexts, it is still time-consuming meaning that the results would hardly be available the same day of a medical consultation i.e., for ANC services. Alternative diagnostic options, such as the point-of-care cathodic circulating antibody (POC-CCA) [33,34] exist but mostly the test is not registered as IVD (in vitro diagnostic) in many endemic countries complicating its use in routine practice outside of research contexts. A POC version of the UCP-LF CAA assay [35,36] is under development, but even if applicability will prove to be successful, its active adoption in clinical practice will require time.

In absence of an easy-to-use diagnostic test and given the high infection rates and the lack of association with any specific factors, our data support MDA for all pregnant women, as recently recommended by WHO. However, more data on safety and tolerability of praziquantel during pregnancy are crucial prior to encouraging this type of policy adaptation.

In our study population, among all women who were offered PZQ treatment, only 0.1% refused. Moreover, we observed a high uptake (30.7%) of preventive treatment (malaria) and (44.1%) supplement uptake (iron and folic acid supplements). In terms of compliance, MDA of pregnant women in highly endemic regions in absence of a clear diagnosis of infection seems feasible.

Several studies have shown the benefit of IPTp programs in Malaria endemic regions [14]. Given that at the current state of the art, PZQ is donated to endemic countries from the WHO without limiting its use exclusively to school age children [7], the integration of PZQ into ANC packages should result in a minor investment for the health system that in exchange could benefit from a high health improvement of their populations. Treatment could be offered within the second trimester following the education sessions that are normally offered during ANC consultations. During these sessions women would have the possibility to ask questions about side effects, safety for the foetus so as get informed about the transmission of schistosomiasis.

Interestingly, our association analysis showed a lower prevalence of infection among those women receiving malaria preventive treatment (51.3% vs. 57.9%, aPR = 0.85, CI 95%: 0.77–0.95). In the past few years, more than one study reported the possible efficacy of antimalarial drugs on schistosome infections [37–39]. The main concern of addressing schistosomiasis treatment with antimalarial drugs is associated to the risk of development of drug resistance for malaria [40], which is a more acute and deadly disease if compared to schistosomiasis. In the highlands of Madagascar, due to altitude and temperatures, malaria does not represent a major health issue since the parasite does not easily circulate [41]. It might be worth to further explore the pharmacological effect of drugs used for malaria IPT against schistosomiasis in order to evaluate the potential of its use in regions such as and similar to the highlands of Madagascar, where malaria does not represent a health concern.

The highly standardized study procedures (data collection and laboratory methods), a precise testing methodology, the continuing training of the staff and the quality assurance measures implemented as well as the large sample size allowed precise inference about the

prevalence of schistosome infection during pregnancy in Madagascar. Despite the strengths listed above, our study does not come without limitations. Our sampling is limited to the highlands of Madagascar hence we cannot generalise the findings to the entire country since in the coastal areas there might be a different dynamic of transmission that could impact on the overall prevalence. Taken into account that data on clinical history were self-reported, we cannot exclude reporting bias for some of the variables discussed (i.e., previous exposure to PZQ). Additionally, in our sample we see a high frequency of women who completed at least primary education which does not reflect the average education level of the country [42]. This might be due to the fact that our recruitment occurred at healthcare facilities where access can be imbalanced in favour of more educated people, showing the potential for some selection bias. Finally, for some of the participants the UCP-LF CAA test results were not available potentially leading to biased estimates. Although we cannot exclude completely this possibility, the sensitivity analysis showed a low risk of bias, supporting robustness of our findings.

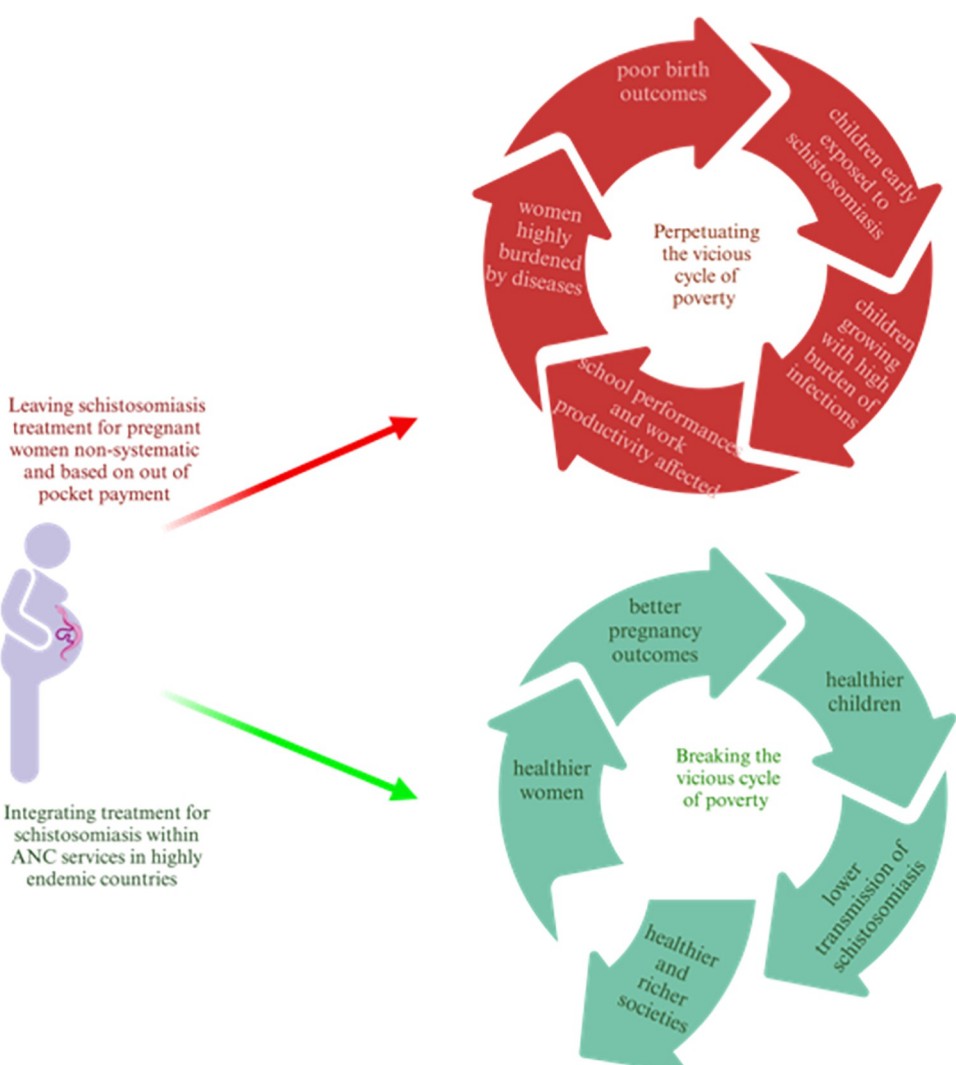

**Fig 3. Theory of change enhanced by integrating schistosomiasis treatment for pregnant women into public health programs.** Created with Biorender.

## Conclusions

In conclusion, our study shows the urgency of treatment for pregnant women in highly endemic context supporting the adaptation of guidelines and policies in those countries where schistosomiasis is shown to be prevalent among adults. The prevalence among adults should be monitored more regularly in endemic countries in order to provide evidence to local stakeholders and policy makers for the need to adapt treatment and preventive strategies for the disease. Based on the findings of the present study, we strongly advocate for adapting schistosomiasis programs with the aim to integrate preventive schistosomiasis treatment in ANCs of highly endemic contexts. This, would contribute to both the WHO and SDGs agendas overall to improving the well-being of women and consequently breaking the vicious cycle of poverty perpetuated by schistosomiasis (Fig 3).

## Supporting information

**S1 Table. Comparison of sociodemographic characteristics of study participants and those excluded from the main analysis due to missing UCP-LF CAA test result.**
(DOCX)

**S1 Dataset. Data used for the analysis.**
(XLS)

**S1 Data Dictionary. Data dictionary used to include the variables in the analysis.**
(XLSX)

## Acknowledgments

We would like to thank the study participants, without whom this work would not be possible, and the technical staff, including drivers, laboratory technicians, study midwives, data clerks and community workers. A special thanks to all the country authorities who allowed and supported the implementation of this study.

The freeBILy consortium composition:
Leiden University Medical Center (LUMC), Netherlands:
Dr. G.J. van Dam
Dr. P.L.A.M. Corstjens
C.J. de Dood
P.T. Hoekstra, MSc
Dr. A.S. Amoah
Dr. M.I. Keshinro
Eberhard Karls Universität Tübingen (EKUT), Germany:
Dr. A. Kreidenweiss
Bernhard-Nocht-Institut für Tropenmedizin (BNITM), Germany:
Dr. N. G. Schwarz
Dr. D. Fusco
Dr. P. Klein
A. Jaeger
Dr. E. Lorenz
Centre de Recherches Médicales de Lambaréné (CERMEL), Gabon:
Dr. A.A. Adegnika
Dr. Y.J. Honkpehedji
Dr. J.C. Dejon-Agobe

R. Beh Mba
M. Mbong Ngwese
M. Nzamba Maloum
A. Nguema Moure
B. Meulah T
Université de Fianarantsoa (UF), Madagascar:
Dr. R. A. Rakotoarivelo
Dr. A. Ralaizandry
Dr. M. Radomanana
Université d'Antananarivo (UA), Madagascar:
Dr. R. Rakotozandrindrainy
Dr. N. Rakotozandrindrainy
Dr. Marie Jeannine Solonirina
Dr. J. Randriamanjara
Centre d'Infectiologie Charles Mérieux (CICM), Madagascar:
Dr. M. Rakoto Andrianarivelo
Dr. T. Rasamoelina
Dr. R. Razafindrakoto
Fundación Privada Instituto de Salud Global Barcelona (ISGLobal), Spain:
Dr. E. Sicuri
C. Aerts, PhD

## Author Contributions

**Conceptualization:** Raphäel Rakotozandrindrainy, Rivo Andry Rakotoarivelo, Tahimandranto Rasamoelina, Philipp Klein, Daniela Fusco.

**Data curation:** Anna Jaeger.

**Formal analysis:** Irina Kislaya.

**Funding acquisition:** Raphäel Rakotozandrindrainy, Rivo Andry Rakotoarivelo, Mala Rakoto Andrianarivelo, Paul L. A. M. Corstjens, Norbert Georg Schwarz, Govert J. van Dam, Jürgen May, Daniela Fusco.

**Investigation:** Valentina Marchese, Tahimandranto Rasamoelina, Jeannine Solonirina, Elveric Fesia Ratiaharison, Njary Rakotozandrindrainy, Mickael Radomanana, Philipp Klein, Daniela Fusco.

**Methodology:** Tahimandranto Rasamoelina, Ravo Razafindrakoto, Nantenaina Matthieu Razafindralava, Eva Lorenz, Pytsje T. Hoekstra, Paul L. A. M. Corstjens, Govert J. van Dam, Daniela Fusco.

**Project administration:** Valentina Marchese, Pytsje T. Hoekstra.

**Supervision:** Raphäel Rakotozandrindrainy, Rivo Andry Rakotoarivelo, Valentina Marchese, Tahimandranto Rasamoelina, Jeannine Solonirina, Elveric Fesia Ratiaharison, Philipp Klein, Norbert Georg Schwarz, Daniela Fusco.

**Validation:** Pytsje T. Hoekstra, Daniela Fusco.

**Visualization:** Daniela Fusco.

**Writing – original draft:** Irina Kislaya, Valentina Marchese, Daniela Fusco.

**Writing – review & editing:** Raphäel Rakotozandrindrainy, Rivo Andry Rakotoarivelo, Irina Kislaya, Valentina Marchese, Tahimandranto Rasamoelina, Jeannine Solonirina, Elveric Fesia Ratiaharison, Ravo Razafindrakoto, Nantenaina Matthieu Razafindralava, Njary Rakotozandrindrainy, Mickael Radomanana, Mala Rakoto Andrianarivelo, Philipp Klein, Eva Lorenz, Anna Jaeger, Pytsje T. Hoekstra, Paul L. A. M. Corstjens, Norbert Georg Schwarz, Govert J. van Dam, Jürgen May, Daniela Fusco.

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
